# Small Split-Ring Resonators as Efficient Antennas for Remote LoRa IOT Systems—A Path to Reduce Physical Interference

**DOI:** 10.3390/s21237779

**Published:** 2021-11-23

**Authors:** Cameron Rohan, Jacques Audet, Adrian Keating

**Affiliations:** Department of Mechanical Engineering, School of Engineering, The University of Western Australia, M050, 35 Stirling Hwy, Crawley 6009, Australia; cameron.rohan4@gmail.com (C.R.); jmaudet6@gmail.com (J.A.)

**Keywords:** planar antenna, PCB antenna, split-ring resonator, metamaterials, LoRa, radio, IOT

## Abstract

While wireless IOT modules can be made extremely compact, antennas typically protrude from the module, providing the potential to catch near moving/rotating equipment or transfer loads to the PCB through end forces, which can lead to failures. This work explores the use of split-ring resonator (SRR) designs to achieve a planar antenna with a maximum dimension less than a monopole working at the same frequency. The very narrow bandwidth of the SRR required detailed physical models to create printed circuit board (PCB)-based antenna designs that could be used at LoRa frequencies of 433 MHz and 915 MHz. Uncertainty analysis allowed for the impact of geometrical and physical tolerances on the resonant frequency to be evaluated. Nearfield and farfield measurements were performed allowing for the resonant frequency, directionality, and range of the antenna to be evaluated. An unbalanced SMA port was added to the SRR design to allow for the use of a network vector analyser to determine the input impedance of various designs. The optimum design achieved an input resistance of 44 Ω at a resonant frequency of 919 MHz, close to the target values (50 Ω at 915 MHz). Field measurements of the received signal strength from a planar antenna design indicated a gain of 5 dB over a conventional quarter-wave monopole antenna, in a footprint that was 40% smaller than the monopole.

## 1. Introduction

Wireless Internet-of-Things (IOT) sensors can be used to monitor or control physical environments that may be in either difficult to reach (remote) or harsh environments [1]. Many of these wireless sensors have a limited amount of power and memory with which to transmit data over a significant distance and are therefore unable to use WiFi or traditional cellular networks [2]. The low-power wide area network (LPWAN) range of protocols and technologies are a solution to these issues, with the long range (LoRa) LPWAN protocol gaining considerable traction.

Beyond these power limitations, many IoT wireless sensors have constraints on their size in order to allow them to have the optimum placement for their application, especially within environments where sealing against weather, sun, and interference from (often rotating) equipment is required. While microcontroller, sensor, and LoRa radio footprints continue to decrease, the main size limiting factor becomes the antenna, which is nominally a quarter wavelength in length. The overall aim of this work is to consider and evaluate how split-ring resonator (SRR) designs can be used to create small, planar antennas for wireless sensor applications. The work presents a resonant frequency model based on the work of Marqués et al. [3,4], which is used to optimize the design of antennas operating at 433 MHz and 915 MHz. These designs are fabricated using conventional Printed Circuit Board (PCB) processes on FR-4, allowing for the experimental evaluation of the resonant frequency and input impedance of several SRR designs. An uncertainty analysis is presented, which identifies key parameters to control for high-quality split-ring resonator antenna designs. Transmission tests over LoRa frequencies of 915 MHz are performed to demonstrate the suitability of SRR antennas, comparing the SRR performance relative a traditional quarter-wave antenna.

### 1.1. Miniature Antennas to Support IOT Systems

The frequency range for LoRa communication is between 400 MHz and 1000 MHz with the specific band dependent on the region; for example, within the Australian context, it is between 915 MHz to 928 MHz [5]. As a result, antenna sizes are approximately 80 mm, which can be significantly larger than the rest of the IoT wireless sensor, making it a limiting factor in the overall size. In addition, traditional monopole antennas are non-planar, extending a considerable way from the main IOT board, presenting an appendage that can catch or interfere with any nearby moving object.

Electrically small antennas can provide a reduced footprint, reducing the chance for the radio system to physically interfere with nearby equipment. An electrically small antenna is defined as having a maximum dimension much less than its wavelength in free space [6]. The common threshold used for this distinction is,
(1)$$a<\frac{\lambda }{2\pi }$$
where λ is the wavelength and a is the radius of the smallest sphere that encompasses the antenna.

A review of miniature antenna structures by Banu and Rathinasabapathy classified a range of physical antenna footprints spanning IOT operating frequencies from 0.868 to 10 GHz [7], showing that the smallest sub-GHz performance was achieved with a non-planar, folded wire structure in front of a ground plane with a footprint of 28 × 10 × 5 mm^3^ [8]. However, in most cases, the miniature antenna structures reported involve rectangular features that can generate multiple resonances. By comparison, split-ring resonators (SRRs) are circular structures that offer a high Q-resonant structure, borrowed from the field of metamaterial research. Metamaterials are artificial composite structures that are built from a lattice of resonators with the aim of altering the electrical properties outside of that which occurs naturally. The electrical size of an SRR is typically in the range of λ/8 to λ/20  of its free space wavelength [4,9,10], with the exact size depending on the design parameters. As an antenna, this structure has the potential to reduce the footprint over traditional quarter-wave monopoles, and being a narrow band resonator, there are potential improvements in gain and efficiency when operated at the centre frequency [11]. The SRR is also easy to fabricate as it is a planar structure that can be manufactured via chemical etching to create PCBs. Traditionally, previous reports have focused on SRR operation greater than 1 GHz [11,12,13,14,15]. However, it is at lower frequencies where the electrical size of traditional antennas can dominate the physical footprint of the system, and where the SRR could provide significant advantage. At these lower frequencies, SRR investigations have experimentally explored radio frequency identification (RFID) applications [10,16] and analysed mobile communication in the GSM bands through simulations [17]. 

While the SRR is a relatively well-understood structure [3,4,18], little research has been conducted on its use as a driven antenna for radio applications. Signal injection has been achieved through coplanar waveguides to inductively couple a signal into arrays of SRR [19] at 7.7 GHz, as well as signal injection via direct connection to the outer ring [10]. However, SRRs have not been used as a directly driven antenna, suitable as a replacement for traditional antennas. In this work, an experimental evaluation of the performance of the SRR as a driven antenna with a particular focus on remote IoT devices using LoRa frequencies of 433 MHz and 915 MHz is undertaken. 

### 1.2. Split-Ring Resonator Overview

The split-ring resonator (SRR) consists of two concentric rings of conducting material, as shown in Figure 1 (blue region) on top of a dielectric, ε. Each ring has a single split, with the splits being located on opposite sides of the structure [3]. The SRR design was first proposed by Pendry et al. [18] in 1999 for use as a metamaterial due to it having an effective magnetic permeability of less than zero when placed in an oscillating magnetic field. Subsequently SRR designs have been proposed for antenna applications. Alici and Ozbay [11] experimentally determined the resonant frequency of a single SRR and its far field radiation pattern. Operation over 9 m at 911 MHz has been demonstrated for RFID tag applications [10] but the performance of the SRR as a long-range antenna in the sub-GHz regime is lacking. SRR have even been shown to offer the ability to steer the emitted radiation, albeit over a small angular range [15]. Beyond these studies, the majority of the research into the SRR has been undertaken in the field of metamaterials, primarily by Marqués et al. [3,20], who studied various properties of the SRR. 

The edge-coupled split ring (referred to in this work as just the split-ring resonator or SRR) originally proposed by Penry [18], resonates via its inherent inductance and capacitance [20]. The capacitance is a result of the gap between the inner and outer rings, while the inductance results from the mutual inductance of the two rings. As found by Marqués et al. [4], at resonance the SRR has both a magnetic and an electric dipole moment. The magnetic dipole moment aligns with the *z*-axis and the electric dipole aligns with the y-axis, where the relevant axes are defined in Figure 1. By making assumptions that the SRR structure is electrically small, the capacitance caused by the splits is negligible and the charge distribution is linear around each ring, the resonator behaviour can be approximated as a simple RLC circuit [3]. This approximate model includes a capacitance caused by the slot line characteristics of the separation and trace width of the two rings; it includes an inductance due to the mutual inductive coupling between the rings and a resistive element due to the no-zero resistivity of the metallic traces. As a result, the resonant frequency is given by:(2)$$\omega_{0}^{2}=\frac{1}{Lp_{hr}C_{pul}/2\;}$$
where $\omega_{0}$ is the resonant frequency, L is the mutual inductance, $C_{pul}\;$ is the capacitance per unit length, and $p_{hr}$ the perimeter of the half-ring. To calculate the capacitance, the same approach can be taken as for a coplanar strip transmission line [21] (see Table 2.7 and Equation 2.4 in [21]) where the per unit length can be determined as outlined in [3]
(3)$$\frac{C_{pul}}{\epsilon_{0}}=\epsilon_{e}\left(\frac{K\left(k^{\prime }\right)}{K\left(k\right)}\right)$$
where K is a complete elliptic function of the first kind and the effective dielectric constant is given by:(4)$$\epsilon_{e}=1+\frac{\epsilon_{r}-1}{2}\left(\frac{K\left(k^{\prime }\right)K\left(k_{1}\right)}{K\left(k\right)K\left(k_{1}^{\prime }\right)}\right)$$
and the geometry-dependent parameters defined in Figure 1 are given by: (5)$$k_{1}=\frac{\sinh\left(\frac{\pi d}{4t}\right)}{\sinh\left(\frac{\pi \left(d+2c\right)}{4t}\right)}$$
(6)$$k=\frac{d}{d+2c}$$
(7)$$\mathrm{and}\;k^{\prime }=\sqrt{1-k^{2}}$$

The inductance of the rings is calculated by [4]: (8)$$L=\frac{\mu_{0}\pi^{3}}{4c^{2}}\int_{0}^{\infty \;}\frac{1}{k^{2}}{\left(bB\left(kb\right)-aB\left(ka\right)\right)}^{2}dk$$
where $\mu_{0}$ is the permeability of free space, c is the trace width, $a=r_{ext}-c/2$ is the average outer radius, $b=r_{ext}-3c/2-d$ is the average inner radius of the SRR, $r_{ext}\;$ is the external radius defined in Figure 1, and B is given by:(9)$$B\left(x\right)=S_{0}\left(x\right)J_{1}\left(x\right)-S_{1}\left(x\right)J_{0}\left(x\right)$$
where $S_{n}\left(x\right)$ is the *n*th order Struve function and $J_{n}\left(x\right)$ is the *n*th order Bessel function. This model indicates the frequency can be tuned by altering the geometric dimensions of the SRR, specifically the average radius $r_{0}=\frac{a+b}{2}=r_{ext}-c-d/2$, ring width c, and ring separation d. The permittivity of the substrate, ε, and substrate thickness, *t*, will also affect the resonant frequency ($\omega_{0})$ but are less convenient to fine tune as they are defined by the PCB manufacturing facility. When calculating the resonance given in Equation (2), the perimeter of the half-ring used in previous works is $p_{hr}=\pi r_{0}\;,\;$ where $r_{0}$ is the average radius (Figure 1) [4]. One potential limitation of this model is that it neglects the contribution of the split-ring gap to the ring capacitance. Inclusion of the effect of the gap has been explored in other work using SRRs with a single ring [22,23], where nearfield material interacting with the gap can be used to alter the resonance. A SRR with dual rings is dominated by the inter-ring capacitance per unit length, $C_{pul}$, but in this work we propose two simple modifications to include the effect of the gap as:
The half-ring perimeter is modified to include the gap by removing the capacitance contribution from the gap by writing the perimeter as $p_{hr}^{\prime }=\pi r_{0}-g$.An additional capacitance $C_{gap}$ associated with both the split-ring gaps is added where:
(10)$$C_{gap}=2\varepsilon_{0}\varepsilon_{r}c*h/g$$
where h is the height of the copper trace, allowing us to re-write the resonant frequency equation as:(11)$$\omega_{0}^{2}=\frac{1}{L\left(\frac{p_{hr}^{\prime }C_{pul}}{2}+C_{gap}\right)\;}$$

The impact of this relatively minor change to the resonance will be evaluated in the work that follows.

Beyond the resonant frequency, the input impedance is also an important characteristic of the SRR, as the highest efficiency for an antenna is achieved when impedance is matched to the source [24]. For most radio frequency (RF) applications, including LoRa antennas, the power source and transmission lines have input impedances of 50 Ω [18] requiring detailed methods to match this impedance of miniature planar antennas [25]. To allow for signal injection to the SRR, Zuffanelli et al. [10] introduced a break at an angle of φ on the outside ring and studied the input impedance as a function of this port position angle (φ) near the resonant frequency. Their results showed an increase in the port resistance as the angle moved from φ = 0°, at the inner split, to φ = 160°, near the outer split, with the impedance closest to 50 Ω near 150°.

The directionality of a SRR antenna is also an important characteristic as it will determine the gain and alignment direction of the corresponding channel antenna [26]. The theoretical direction of the magnetic and electric dipole moments has been discussed by Marqués et al. [20] with the electric dipole in the y-direction of Figure 1 and the magnetic dipole in the z-direction. Alici and Ozbay [11] considered the nearfield angular radiation patterns from SRR antennas directly above a ground plane. The angular dependence of SRR antennas from nearfield measurements above a ground plane was also measured in the current work to compliment the findings of the farfield range tests, which did not include a ground plane.

## 2. Methods

### 2.1. Model Implementation

The analytical SRR model from Marqués et al. [4], detailed through Equations (2)–(11), was evaluated using a Python script to calculate the resonant frequency of an SRR given the geometric parameters. This required the inductance and capacitance of the SRR to be determined, however, the results were found to be particularly sensitive to the value of capacitance chosen. The model used a number of parameters to determine the design of the SRR antenna as indicated in Figure 1: average radius, $r_{0}$; external radius,$\;r_{ext}$; ring trace width, c; the separation between the inner and outer ring, d; split gap, g; and dielectric thickness, t. The dielectric thickness is determined by the FR4 substrate onto which the antenna is printed. The SRR antennas were manufactured by a commercial PCB manufacturing house (JLCPCB) using an FR4 dielectric thickness of t= 1.6 mm, a dielectric constant of ε = 4.4, and copper thickness of h=35 μm. In the model defined in Equations (2)–(11), the split gap is assumed to have negligible effect on the resonant frequency. Preliminary designs used just three parameters to adjust: external radius, $\;r_{ext}$; ring trace width, c; and ring separation, d. The preliminary designs were based on the physical characteristics of the 911 MHz passive RFID designs from Zuffanelli et al. [10], utilising their radius to trace width, scaled to our operating frequencies. For the evaluations in this paper, two frequencies of interest were chosen, 433 MHz and 915 MHz corresponding to the licence-free radio frequency bands that are commonly used for LoRa transceivers.

### 2.2. Antenna Fabrication

Two types of SRR were fabricated: preliminary designs consisted of traditional split ring, whose parameters are defined in Figure 1 and described through the models of Equations (2)–(11). The second set of designs included a cut in the outer ring to allow for the addition of a physical SMA connector for signal injection. The preliminary fabricated PCB SRRs are shown in Figure 2a. These preliminary antenna designs were divided into two groups, with five designs targeted for 433 MHz operation and five designs aimed at 915 MHz. Both sets of designs follow the same design rules. Table 1 details the parameters used in all designs. Design 1 is aimed as close as possible to the target frequency, either 915 or 433 MHz, with reasonable PCB fabrication values for the trace width and ring separation. In each frequency band, Designs 1–3 all have the same trace width and ring separation, while the external radius is altered to shift the resonant frequency by ±10%. Design 2 at a frequency 10% higher, and Design 3 is aimed at a 10% lower resonant frequency. The parameters of Design 4 were chosen to minimise the overall size of the ring, and finally Design 5 was chosen to maximise the trace width. These designs were chosen after an exploration of the parameter space, by perturbing the physical dimensions used in the model created from Equations (2)–(11) to achieve the desired resonance in Equation (2). Comparing the ratio of SRR diameter to the equivalent λ/4-monopole length shown in Table 1, the ratios range from 0.25 to 0.39, indicating all designs were 40% smaller than an equivalent λ/4 monopole. These results are consistent with previous electrical small antenna designs of around $\frac{\lambda }{10}$ [11], however, we feel a comparison with a $\frac{\lambda }{4}$ monopole is more relevant to demonstrate enhancements over traditional antennas.

### 2.3. Characterisation

Prior to designing the second set of SRR antennas with an outer ring containing a physical port (SMA connector), a non-contact measurement of the preliminary designs was performed by coupling the SRR to a nearfield monopole. This allowed for the designs of the fabricated structures to be compared against the analytical model Equation (2).

For resonant frequency measurements, a non-contact experiment was used similar to that by Alici and Ozbay [11]. Here, a monopole antenna was used as the transmitter/receiver to measure S_11_ (signal reflections) interacting with the SRR in the nearfield regime. The setup consisted of a monopole antenna mounted to a 280 mm × 255 mm copper ground plane, as shown in Figure 2b. These tests evaluated the LoRa designs at 433 MHz and 915 MHz detailed in Table 1. The monopole was constructed from the core of a 50 ohm coaxial cable stripped down to the dielectric. The inner, solid copper conductor had a diameter of 1.0 mm, and the polyethylene dielectric had an outer diameter of 3.0 mm. As there are SRRs designed for both 433 MHz and 915 MHz, different monopole antenna lengths were used for each, namely 100 mm and 50 mm, respectively. The length of the monopole antenna was intentionally chosen so that the monopole’s own centre frequency did not interfere with the SRRs centre frequencies but still provided sufficient radiation to excite the SRR. By placing the monopole’s centre frequency 20–30% above the SRRs’ resonant frequency, higher-order radiation patterns from the monopole were avoided. A reference (baseline) reflection measurement of the spectrum was performed to obtain $S_{11}^{ref}$ for the monopole in the absence of the SRR. Subsequently, spectral measurements of the monopole were performed with the SRR in place (in various orientations) resulting in the measurement $S_{11}^{SRR}$. To drive the monopole antenna, the output port of a two-port vector network analyser (DG8SAQ VNWA-3 Low Cost 1.3 GHz Vector Network Analyser) was used. When under test, the SRR was held in place with formed polystyrene, which had a negligible effect on the measurement. Custom cut polystyrene allowed for the rotation of the SRR about all three axes defined in Figure 1 and is shown rotating about the $\theta_{x}$ direction in Figure 2b.

The second set of designs for the PCB SRR were optimized for 915 MHz and included ports at various angles, introduced into the outer ring of the SRR onto which an SMA connector could be attached. This allowed for the determination of the optimum port angle at which the best antenna matching condition could be found. Once found, the design with the best match was used for range tests, with the signal injection via a standard RF-SMA connector into the port of the SRR antenna. Range tests were performed at 915 MHz within an open field at a nominal height above the ground of 1 m.

## 3. Results

### 3.1. Characterisation of Preliminary Designs

Table 2 shows both the calculated resonant frequencies from the numerical model described through Equations (2)–(11) and our experimentally measured resonant frequencies for the preliminary designs shown in Figure 2a. Resonant frequencies were measured using the nearfield S_11_ coupling from a monopole to the SRR, relative to the monopole’s response alone. The predicted frequencies obtained from Equation (2) typically overestimated the measured resonant frequency by an average of 5% for the 433 MHz designs and 2.7% for the 915 MHz designs. Importantly, the root-mean-squared (rms) variation in $f_{0}$ was less than 4% (for 915 MHz designs), indicating a low variation in the predicted values.

For 915 MHz designs, Figure 3 shows the results of the model and experimental comparisons due to geometric parameter variation where the permittivity for the FR4 substrate ε=4.4 and the other parameters are as indicated in the figure inset. The experimentally measured resonant frequencies show an increase in resonant frequency as the split-ring gap (g) increases, as shown in Figure 3a. However, models that do not consider the effect of the gap (indicated by model without the gap label) do not show this variation. The modified models proposed in Equation (11) show good agreement with the measurements, indicating the primary effect the gap on the resonant frequency was captured. The effect of the trace width (c) on the resonant frequency is considerably greater than the effect of split-ring gap, as shown in Figure 3b. Here the inclusion of the gap in the model tends to increase the resonant frequency by only 15 ± 1.5 MHz. These results indicate that, in all cases, for a design frequency of 915 MHz, the experimental results were within 6% of the model predictions.

### 3.2. Uncertainty Analysis of Resonant Frequency in SRR Designs

As the model contains multiple parameters, understanding the relative importance of each parameter on the accuracy would help the overall design process. Here, we used uncertainty analysis methods [27] to determine the influence of each parameter. Figure 4a shows the calculated sensitivity of the resonant frequency due to each physical parameter. In analysing the effect of each parameter, it is assumed the planar dimensional tolerances are ±0.1 mm, the copper thickness tolerance is ±5 μm, and the dielectric constant tolerance is ±0.2. The impact of these variations to the uncertainty in the resonant frequency is shown in Figure 4a and expressed as a tornado diagram, shown in Figure 4b, where the impact of each uncertainty was ordered relative to its importance. From this analysis, it appears that the separation (d) between the inner and outer ring had the largest impact on the resonant frequency—the tolerance of the separation introduced the largest variation of ±56 MHz on the resonant frequency.

### 3.3. Orientation Dependence of SRR Antennas

Figure 5 shows the nearfield interaction of a monopole with SRR antenna designs ($S_{11}^{SRR}-S_{11}^{ref}$) optimized for 433 MHz LoRa frequencies under various orientations. In Figure 5a,c, the x-axis (defined in Figure 1) of the SRR antenna is parallel to the monopole. When rotating about the x-axis, the initial $\theta_{x}=0$ position is defined when the plane of the PCB is aligned to the monopole. In this position, the y-axis is perpendicular and pointing towards the monopole. The peak of the S_11_ coefficient of the monopole indicates the degree of interaction (coupling) between the SRR and the monopole shown in Figure 5b. As the magnetic field rotates around the monopole, maximum coupling is expected and observed when $\theta_{x}$ = 0 or 180° when the z-axis of the SRR is maximally aligned with the magnetic field. The symmetrical response measured for this alignment is expected.

Figure 5c,d show the coupling when rotated about the y-axis $\theta_{y},$ and $\theta_{x}=0$ and $\;\theta_{z}=0$. When the x-axis of the SRR is parallel to the monopole, $\theta_{y}$ = 0. The coupling to the SRR was 0.6 dB greater when the split in the outer ring was pointing upwards ($\theta_{y}$ = 0°) compared to facing towards the ground plane ($\theta_{y}$ = 180°). This may indicate that the magnetic field is not uniform along the length of the monopole antenna in the nearfield.

With the xy-plane aligned to the monopole, rotation about the z-axis $\theta_{z}$ (perpendicular to the monopole) when $\theta_{x}=0$ and $\;\theta_{y}=0$ resulted in a sinewave-like variation in the coupled power, as shown in Figure 5e. We observed that each of the datasets resulted from a different measurement, and while every care was taken, the $\theta_{z}$ = 0 value in Figure 5e differs by 0.15 dB compared with Figure 5b,d, largely due to the frequency resolution of the VNA, control of the SRR-to-monopole spacing, and the effect of the normalisation. Nevertheless, this result indicates the maximum signal response is expected when $\theta_{z}=0$, $\theta_{x}=0,$ and $\;\theta_{y}=0$ and, hence, this represents the optimum orientation required for efficient signal propagation required for our subsequent range tests. The results of Figure 5 clearly indicate a strong dependence on orientation relative to the monopole, indicating a directionality of the antenna, consistent with previous work [4,10].

### 3.4. Characterisation of Driven Designs with Ports Added

While monopole–SRR nearfield coupling assists with characterisation of the resonant frequency of the antenna, it does not allow for transmission using the SRR. To allow for driving of the SRR, the PCB was modified to include a break, as shown in Figure 6a, which allowed for the addition of a standard SMA connector, as shown in Figure 6b.

The effect of the angle φ of the SMA port connected to the outer ring was investigated in Figure 7a, which shows that increasing the port angle resulted in an increase of the input resistance (right y-axis) of the SRR at the resonant frequency. All of the input resistances measured were below the desired 50 Ω, with the highest resistance of 44.3 Ω occurring at a port angle of φ = 163°. Figure 7a also shows that the S_11_ reflection coefficient (left y-axis) observed at the input to the SRR decreased significantly as the port angle increased, with the lowest S_11_ of −24.4 dB. The φ = 0° port angle was omitted from the graph as no definite resonance was observed across the spectra at this angle. The trend of increasing input resistances matches that found by Zuffanelli et al. [10]. Representative spectra at φ = 163° and φ = 90° are shown in Figure 7b, which show a clear, strong resonance at φ = 163°, which fit well to the Lorentzian spectra with a half-width of δf=60 MHz and a centre frequency of 912 MHz. This indicates the addition of the SMA port to the SRR design did not significantly alter the design frequency of 915 MHz. However, the narrow bandwidth of these SRR antennas suggests that fabrication tolerances discussed in relation to Figure 4 need to be considered to avoid the fabricated designs falling out of the band.

### 3.5. LoRa Testing of SRR Antennas

We performed open-air range tests to allow researchers the chance to compare our results with models and experiments [28] relevant to remote operations using LoRa for IOT, as opposed to the use of a large-scale Anechoic chamber. Each measurement is the average of five received signal strength (RSS) measurements as measured by the LoRa receiver. The SRR was oriented as previously described (Figure 5) with the z-direction horizontal, and xy-plane aligned to the monopole $(\theta_{x}=0{}^{\circ},\;\theta_{y}=0{}^{\circ},\theta_{z}=0{}^{\circ}$). In the measurements presented, the farfield is defined at distances beyond 20λ. Models and measurements of LoRa transmission systems with a ¼-wave monopole suggest that line-of-sight transmission ranges of up to 15 km can be achieved [28]. In the farfield arrangement shown in Figure 8a, the power used in this experiment was set to the default setting of +14 dBm [5] to allow for the measurement of the RSS within a measurement distance of less than 250 m.

Figure 8b shows field measurements of the RSS at 915 MHz comparing two different SRR designs and a traditional quarter-wave monopole. The measured distance (top x-axis) was also plotted as a function of the inverse squared distance (bottom x-axis) to highlight the expected variation of the radiated power in the farfield—the near linear variation shows that the propagation from the SRR is similar to a standard monopole antenna. The sensitivity limit of the LoRA receiver was −111 dBm [29], which occurred at a distance of around 1 km from Figure 8b. In the open-air experiment conducted, the nearest reflecting object (tree) was around 100 m away, potentially introducing interference at −80 dBm to −90 dBm in the RSS. In this measured range, a small deviation from linearity on the $1/r^{2}$ axis appears, possibly due to such reflections. A jack-knife uncertainty analysis [30] of the linear fit (m*x*+$y_{0}$) to the data in Figure 8b revealed an uncertainty in the RSS offset ($y_{0}$) for the SRR of ±0.13 dB, while for the monopole, the uncertainty was ±1.1 dB (with 95% confidence). The port angle directly affects antenna matching and hence transmitted power–port matching, which is close to 50 Ω and improves the achievable transmission range. Of the SSR designs, the one with the port angle at 163° showed the highest signal strength across all ranges tested, showing more than 5.3 dB improvement over the quarter-wave monopole, with an uncertainty of ±1.1 dB. Importantly, the 90° SRR produced the worse signal strength of any antenna evaluated. These results are consistent with both the nearfield measurements of Figure 5 and the matching performance analysed in Figure 7 after connection of the SMA port. The results confirm the improvement, which can be obtained with correctly designed SRR antennas for practical use in IOT-based radio systems.

## 4. Discussion

While the addition of the SRR introduces directionality, which might be considered to complicate device orientation and placement, the benefits of the small footprint and efficient power utilisation within a low-power network are considered worthwhile compromises. These antennas are considerably smaller than the equivalent monopole types typically used for IOT devices, with the largest dimension at only 40% of that of a monopole antenna. Preliminary calculations based on scaling the results of Figure 3 to WiFi frequencies at 2.5 GHz suggest a SRR radius of only $r_{ext}$ = 6.6 mm is required. However, fabrication tolerances discussed in Figure 4 would have considerably more impact at these higher frequencies, requiring higher tolerance on the PCB fabrication or a method to tune the designs. One possible method to tune the centre frequency is suggested by the data in Figure 4a, which shows the sensitivity to the relative dielectric constant is $\partial f_{0}/\partial \varepsilon_{r}$ = −83 MHz and the thickness is $\partial f_{0}/\partial t$ = −47 MHz/mm. Hence, increasing the dielectric or thickness (by the addition of stacked substrates) could potentially lower the resonant frequency, while lowering $\varepsilon_{r}$ or t (by removing portions of the substrate or polish-back) could be used to increase the resonant frequency. These methods could be used to tune antennas to achieve the desired performance where fabrication tolerances are too large or too costly to achieve.

The gain achieved from our measurements shown in Figure 8b exceeds that of previous miniature antenna designs in the sub-GHz band [7]. The antenna gain largely achieved results from the directionality observed in Figure 5 from these SRR metamaterial designs. Traditional monopole antennas radiate isotopically, which can waste considerable power, and their use assumes no network topology awareness, which is often not the case. Where the location of the nodes and network gateway are known and fixed, this knowledge can be utilized to more efficiently use power radiated from the IOT radio antenna. Comparable directional antenna designs include Yagis [31] and SPIDA [32] designs, both of which increase the total physical dimensions to at least a half-wavelength. However, we propose that electronically steerable antenna designs such as the SPIDA could leverage the SRR design to achieve a reconfigurable, small footprint antenna. The closest comparable work to date with a matched port on a SRR achieved reception at 911 MHz over only 9.3 m [10]. Given the size and performance benefits, these SRR antenna designs provide significant improvement when using IOT devices in harsh and remote environments, allowing for a reduction in power consumption and elimination of points of interference, which can arise from protruding antennas structures. These power consumption gains are particularly important in a low-power LoRa system, typically used in remote locations where efficient power utilisation is the key to providing a low maintenance, reliable data connection.

## 5. Conclusions

This work has reviewed the basic design process for creating SRR antennas on a planar, PCB platform. Designs at 433 MHz and 915 MHz were evaluated showing excellent match to theoretical predictions. Such models are extremely important for SRR structures given their very narrow resonant bandwidth. An uncertainty analysis was performed to identify the effect of various geometric and physical parameters on the resonant frequency. Measurements of the input characteristics of SRRs with different port angles were also performed. An increasing port angle was shown to produce an increase in input resistance, with the best impedance match in the devices fabricated achieved at a port angle of 163°, which produced an input resistance of 44.3 Ω and a reflection coefficient of −24.4 dB. Nearfield and farfield results were extremely consistent, and indicated the addition of the port to the SRR design did not significantly alter the SRR resonant frequency. Experimental field measurements of SRR antennas for LoRa applications were undertaken to determine the range of the transmission, with more than 5 dB improvement observed over traditional quarter-wave monopole structures. These results suggests that the SRR antennas have the potential to be useful in harsh wireless sensor applications where small size and non-protruding features are key operational requirements.

## Figures and Tables

**Figure 1 sensors-21-07779-f001:**
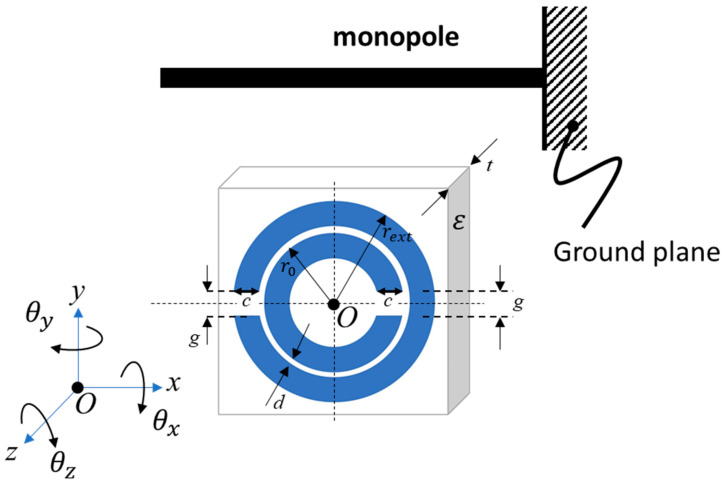
Diagram of a split-ring resonator (SRR) with the dimensions and coordinate system defined consistently with the results presented in this work. A monopole antenna oriented parallel to the x-axis is shown, which is used to excite the SRR during rotations $\theta_{x}$, $\theta_{y},$ and $\theta_{z}$ (as shown $\theta_{x}=0{}^{\circ}$, $\theta_{y}=0{}^{\circ},$ and $\theta_{z}=0{}^{\circ})$. Observations are made by measuring its effect on the S_11_ parameter of the monopole antenna’s port. The defined parameters are $r_{0}$, the average radius, $r_{ext},$ the outer ring radius, *c*, the ring width, *g*, the split-ring gap, *t*, the substrate thickness, and *d*, the separation between rings.

**Figure 2 sensors-21-07779-f002:**
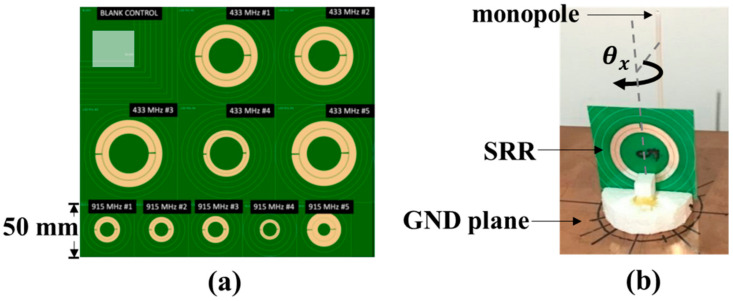
(**a**) Preliminary layout of SRR designs at 433 MHz and 915 MHz fabricated by standard manufacturing processes; (**b**) experimental setup to measure S_11_ coupling from monopole to SRR, while the orientation of the SRR is varied. Orientation $\theta_{x}=90{}^{\circ},\;\theta_{y}=0{}^{\circ},$ and $\theta_{y}=0{}^{\circ}$ is shown.

**Figure 3 sensors-21-07779-f003:**
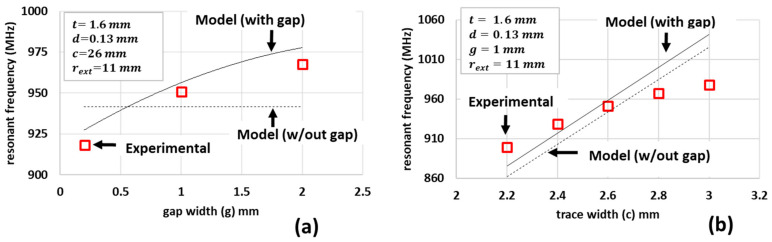
Measured and calculated resonant frequencies of SRRs due to changes in (**a**) gap width $\left(g\right)$ and (**b**) trace width. All other parameters shown were held constant.

**Figure 4 sensors-21-07779-f004:**
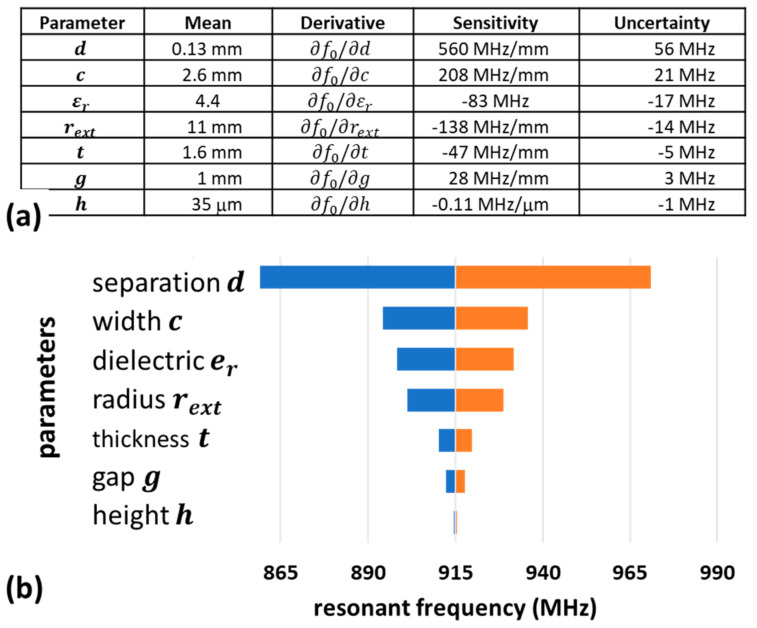
(**a**) Table of parameters and the calculated sensitivity of each parameter (derivatives) on the resonant frequency. The calculated uncertainty assumes ±0.1 mm maximum tolerance variation in the PCB geometry, +/−5 μm in the copper thickness, and ±0.2 variation in the dielectric constant. (**b**) Tornado diagram representation of the impact the uncertainties in (**a**) have on the calculated resonant frequency.

**Figure 5 sensors-21-07779-f005:**
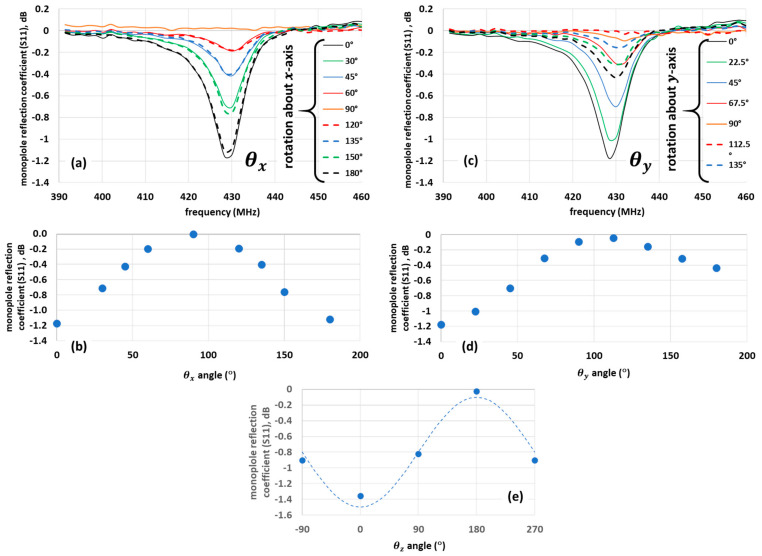
The effect of rotation on the reflection coefficient measured by the monopole interaction with nearfield SRR antenna. The x-axis is parallel to the monopole, with the outer split upward and in plane of the SRR at $\theta_{x}=0$, $\theta_{y}=0$,$\;and\;\theta_{z}=0$ —see Figure 1. Spectra from a 433 MHz antenna design when rotated (**a**) about the x-axis $\theta_{x}$ ($\theta_{y}=0$,$\;\theta_{z}=0$ ) and (**b**) the peak reflection coefficient rotated about $\theta_{x}$. (**c**) Spectra when rotated about the y-axis $\theta_{y}$ ($\theta_{x}=0$,$\;\theta_{z}=0$ ) and (**d**) the peak reflection rotated about $\theta_{y}$. (**e**) The peak reflection value rotated about $\theta_{z}$ ($\theta_{x}=0$,$\;\theta_{y}=0$ ) with the xy-plane aligned to the monopole (z-axis normal to the monopole). The general fit is overlaid as a dash line.

**Figure 6 sensors-21-07779-f006:**
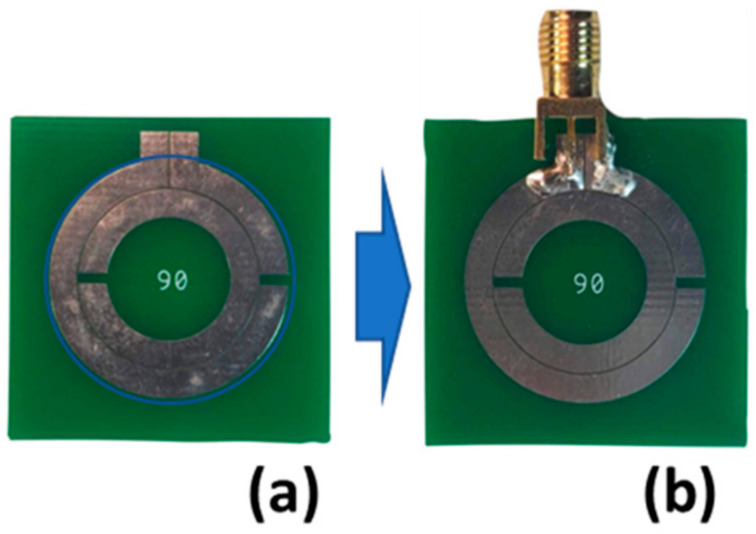
SRR with port added by introducing a break in the outer ring: (**a**) bare input port at 90° and (**b**) input port at 90° with SMA connector to drive the SRR antenna.

**Figure 7 sensors-21-07779-f007:**
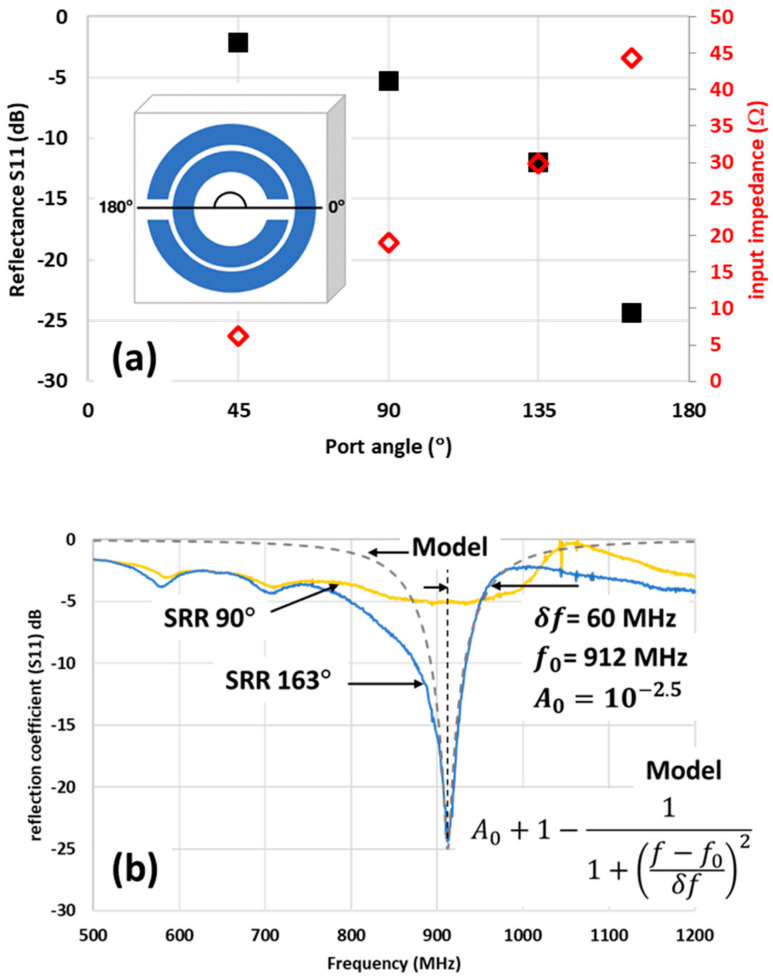
(**a**) Variation of (left axis) reflectance S_11_ parameter and (right axis) input impedance at the resonant frequency for different port angles. The port angles are defined in the inset shown; (**b**) representative S_11_ spectra at port angles of φ = 163° and φ = 90°.

**Figure 8 sensors-21-07779-f008:**
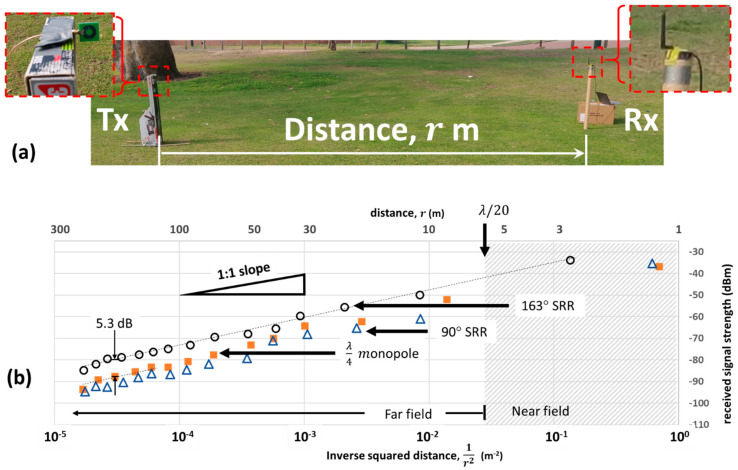
(**a**) Experimental arrangement for measurement of the range variation of received signal strength (RSS). The SRR xy-plane is aligned with the monopole with ($\theta_{x}=0{}^{\circ},\;\theta_{y}=0{}^{\circ}$); (**b**) RSS values recorded with different sending antennas (SRR 90° port △; SRR 163° port ⚪; monopole 
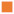
) over a range of distances. Secondary x-axis shows variation in the inverse squared distance. The SRR with a 163° port angle shows a 5.3 dB improvement in gain over the λ/4 monopole in the farfield.

**Table 1 sensors-21-07779-t001:** Physical dimensions of the SRR PCB designs and the resulting modelled frequency expected from each design. Nominally designs at each wavelength were perturbed ±10% of the desired frequency of 433 MHz and 915 MHz to allow for the fabrication and modelling uncertainties.

	433 MHz Design Series					915 MHz Design Series				
Design	1	2	3	4	5	1	2	3	4	5
Modelled frequency (*f*_0_)	437.5	481.2	393.7	440.5	439.7	927.3	972.5	880.6	1002.1	917.1
external radius (*r_ext_*)	28.9	27.15	31.02	21.40	29.95	12.00	11.63	12.42	9.41	15.85
trace width (*c*)	6.20	6.20	6.20	3.00	6.92	2.60	2.60	2.60	1.50	5.00
separation (*d*)	0.31	0.31	0.31	0.13	0.31	0.13	0.13	0.13	0.13	0.13
mean radius (*r*_0_)	22.55	20.80	24.67	18.34	22.88	9.34	8.97	9.76	7.85	10.79
gap (*g*)	1.00	1.00	1.00	1.00	1.00	0.50	0.50	0.50	0.50	0.50
Diameter: λ/4 monopole	0.34	0.35	0.33	0.25	0.35	0.30	0.30	0.29	0.25	0.39

**Table 2 sensors-21-07779-t002:** Comparison of experimental and modelled (predicted) resonant frequency values for LoRa designs nominally at 433 MHz and 915 MHz. Prediction error shown indicating a rms error of 3.6% at 915 MHz and 1.3% at 433 MHz.

433 MHz Series	SRR Design 1	SRR Design 2	SRR Design 3	SRR Design 4	SRR Design 5	rms	mean
Experimental	413.6	457	373.4	427.8	413.9	**30.2**	**417.1**
Predicted	437.5	481.2	393.7	440.5	439.7	**31.0**	**438.5**
Prediction error	5.78%	5.30%	5.44%	2.96%	6.23%	**1.27%**	**5.14%**
**915 MHz Series**	**SRR Design 1**	**SRR Design 2**	**SRR Design 3**	**SRR Design 4**	**SRR Design 5**	**rms**	**mean**
Experimental	904.6	938.2	851.5	1032.4	856.8	**73.9**	**916.7**
Predicted	927.3	972.5	880.6	1002.1	917.1	**47.8**	**939.9**
Prediction error	2.51%	3.65%	3.41%	−2.94%	7.04%	**3.61%**	**2.74%**

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
