# Peer review of "Small Split-Ring Resonators as Efficient Antennas for Remote LoRa IOT Systems—A Path to Reduce Physical Interference"

_sensors, 2021, doi:10.3390/s21237779_

Round 1

Reviewer 1 Report

The work presents the design of Planar Split-Ring Resonator Antennas for LoRa based IoT systems. The design, fabrication, implementation, calibration and demonstration have been presented. The predicted and the measure values are much in agreement with error % maintained around 5% and in many cases less than that. It would be good i fthe authors give more discussion on the effect of the angle on the freuency response of the antenna. What is the maximum range of the proposed LoRa in the expreriment? What is the effect of the angle of port on the overall systems performance in the context of the placement of the system? Is it not complicating the orientation of the device and its placement? Is it possible to mitigate this?

Reviewer 2 Report

The article introduces an interesting job. In the software and hardware design to the algorithm are involved. The article has done the relative contrast design experiment. The article is of some significance in general and some suggestions are expected to be considered.

  1. It is suggested that the title of the article be adjusted. Adjust according to the Special Issue theme.
  2. Introduction in the first part of this article suggests further arrangement. This section focuses on the content of the article, some of which can be put into the method introduction later.
  3. There are a lot of new technologies, such as antenna array technology, etc. I suggest you take a look at them.
  4. It is suggested that the structure of the article be greatly adjusted. It would be helpful to look at the structure of other articles in the SENSORS journal.

Reviewer 3 Report

In this work a planar antenna design based on a Split Ring Resonator (SRR) properly frequency tuned for LoRa transceivers (433MHz or 915MHz) is presented and evaluated. The presented antenna has a size less than 40% of the corresponding length of a λ/4 monopole antenna.

General comment:

The idea of using the SRR design as a single radiator instead of using it as unit cell in metamaterial surfaces is known and there are several works that theoretically analyze this design and experimentally verify the theoretical results. The authors claim that this design has not adequately examined experimentally in LoRa frequencies (433MHz and 915MHz). In my opinion this statement is valid just for lower band therefore I am very skeptical about the novelty of the propose design.

Detail Comments, Typos and Language corrections:

Line 59-line 65: Please rephrase the sentences (However…However…However…)

Line 78: fabricated (instead of fabrication)

Line: 93: MHz (instead of Hz)

Fig.1: Line 101: A monopole (instead “An”) and “duration” (during?)

For the consistency of the text please give the definition of the parameter ro (average of ?) because in the text there are various terms: e.g. “ ?o, the average of inner and ?ext outer ring radius”, “average radius ?o,…”, “where ?o is the average radius of the inner and outer rings…”, “mean radius ro”(Table 1)

For the same reason in Line 130: give the relation between parameters a and b with ro

Line 213: correct reference [10] to [5]

Line 229-230: difficult to understand, please rephrase the sentence.

In Methods section: The authors should give more information about the used algorithm (optimization) in order to derive the dimensions given in Table 1.

In Results section (Line 243- ), for the statement: ”The prediction typically overestimates the resonant frequency by an average of 5% for the 433MHz designs and 2.7% for the 915 MHz designs” please provide a reference

Regarding Antenna evaluation measurements: The procedure that has been followed in order to evaluate the antenna performance was not the conventional one:

a) The authors should explain why they did not simply use the method of measuring the S11 using a VNA port1 connected at SMA connector of each one of the ten (10) antenna prototypes  in order to get the results of Table 2.

b)The authors should also explain why the radiation diagram of the antenna did not measured in a far-field test site (Anechoic chamber), instead they prefer to use a measurement campaign in an open area test site (fig 8). Please comment on the measurement accuracy (interference on the results caused by reflections) and uncertainty that the used method of measurements has.

The authors should explain the worth of the measurement results included in fig 5 regarding the characterization of the proposed antenna. They just prove the already known theoretical and experimental results from previous publications (e.g. [4], [5])

Line 329: overlaid (instead “overaid”)

Line 334: Please correct fig 5d to 5e, and explain why in fig5e at θz=0deg the value of S11 is not ‑1.2dB as in fig5b and fig5d

From fig7a it seems that measurements for angle φ larger than 163 deg were not taken. Then, how you conclude that the “highest resistance of 44.3 Ω occurring at a port angle of φ =163deg”?

Round 2

Reviewer 2 Report

Part 3 is a bit too much and not so coordinated. Please discuss it again. The authors decide whether to adjust this part. If the authors think no adjustments are needed, that's fine.

Please check the language and format carefully.

Author Response

Edits to section 3 have been made in various parts (detailed with track changes in the manuscript) which has reduced that section by 13% without compromising understanding or the core contributions.  We hope this addresses the concerns from this eviewer. Edits (showing deleted sections) atatched.

The manuscript has been re-read and edits made to correct for langauge

Reviewer 3 Report

Most of my comments have been sufficiently addressed in the new version of the manuscript. I think that the work can be accepted for publication.

Author Response

minor edits made to english langauge.

We thank the reviewer for their comments